# Using delayed decoupling to attenuate residual signals in editing filters

Kenneth A. Marincin[1], Indrani Pal[1,†], Dominique P. Frueh[1]

[1]Department of Biophysics and Biophysical Chemistry, Johns Hopkins School of Medicine, Baltimore, MD, 21205, USA
5   †Current address: Department of Biochemistry and Molecular Biophysics, Kansas State University

*Correspondence to*: Dominique P. Frueh (dfrueh1@jhmi.edu)

**Abstract.** Isotope filtering methods are instrumental in biomolecular nuclear magnetic resonance (NMR) studies as they isolate signals of chemical moieties of interest within complex molecular assemblies. However, isotope filters suppress undesired signals of isotopically enriched molecules through scalar couplings, and variations in scalar couplings lead to imperfect suppressions, as occurs for aliphatic and aromatic moieties in proteins. Here, we show that signals that have escaped traditional filters can be attenuated with mitigated sensitivity losses for the desired signals of unlabeled moieties. The method uses a shared evolution between the detection and preceding preparation period to establish non-observable antiphase coherences and eliminates them through composite pulse decoupling. We demonstrate the method by isolating signals of an unlabeled post-translational modification tethered to an isotopically enriched protein.

## 1 Introduction

Nuclear magnetic resonance (NMR) has become a mainstay of biomolecular studies, notably because its non-invasive nature makes it particularly suited to study interactions between biomolecules at the atomic level, such as protein-protein (Sprangers and Kay, 2007), protein-DNA/RNA (Kalodimos et al., 2004), and protein-small molecule interactions (Meyer and Peters, 2003). A key to this success is the ability to control the isotopic labelling of the molecules participating in the interactions and use spin dynamics to isolate signals of interest in otherwise crowded spectra. Thus, in mixtures of isotopically labeled ($^{15}$N/$^{13}$C) and unlabeled ($^{14}$N/$^{12}$C) components, signals from the labeled components can either be selected (isotopic editing) or eliminated (isotopic filtering) (Otting and Wüthrich, 1990). Imperfections in isotopic filtering of labeled signals is recognized as a common challenge and unsuppressed signals can bias interpretations of results. We have been motivated to overcome this challenge within the framework of our studies of nonribosomal peptide synthetases (NRPSs), a family of enzymatic systems that produce important pharmaceuticals such as antibiotics (bacitracin, vancomycin), anticancer agents (epothilones), or immunosuppressants (cyclosporine) (Finking and Marahiel, 2004; Süssmuth and Mainz, 2017).

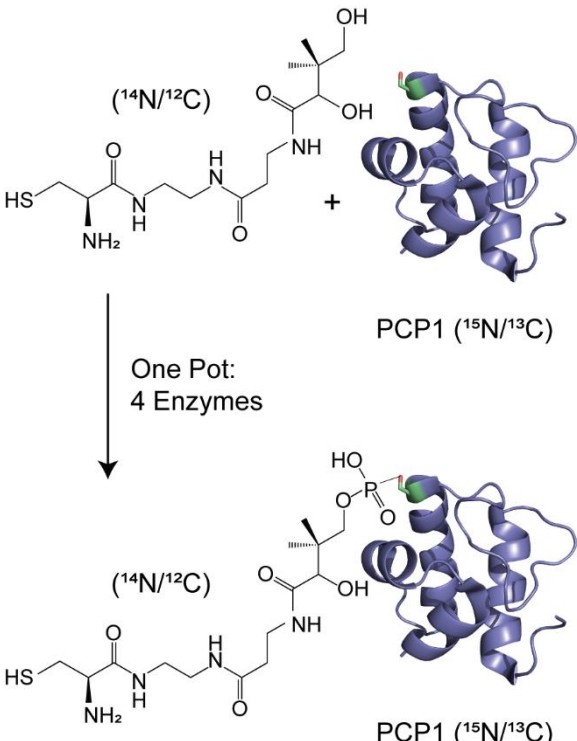

**Figure 1. Isotopic labelling scheme used in our studies.** Isotopically enriched PCP1 ($^{15}$N/$^{13}$C) is chemoenzymatically tethered with an unlabeled pantetheine analogue ($^{14}$N/$^{12}$C) harboring a cysteine substrate ($^{14}$N/$^{12}$C). See Sect. 3.2 for more details. PCP1 solution structure: PDB 5U3H (Harden and Frueh, 2017). Analogue structures made in ChemDraw (PerkinElmer Informatics).

Isotopic filtering will enable the study of interactions between NRPSs and their substrates. NRPSs employ domains called carrier proteins (CPs) to covalently tether substrates and shuttle them between partner catalytic domains. These domains are organized in contiguous modules, and substrates attached to CPs on sequential modules are condensed such that an upstream CP donates its substrate to a downstream CP, which then harbors an extended intermediate. The process is iterated until product formation. In this, carrier proteins are covalently modified with a 20 Å long phosphopantetheine (PP) group (Lambalot et al., 1996) which gets covalently loaded with an amino acid substrate. We and others have found that some CPs interact transiently with their tethered substrates (Goodrich et al., 2015; Jaremko et al., 2015) such that the phosphopantetheine group and its attached substrate sample both an undocked state and a docked state. The finding is significant as the modular architecture of NRPSs is suitable for producing new pharmaceuticals by engineering NRPSs to incorporate new substrates, and these transient interactions may need to be preserved in artificial NRPSs. In the current work, we focus on the 9.6 kDa carrier protein PCP1 isolated from yersiniabactin synthetase, which natively harbors a cysteine substrate (Gehring et al., 1998). In order to study interactions between PCP1 and the phosphopantetheine group and its attached cysteine substrate, we have chemoenzymatically (Kittilä and Cryle, 2018; Worthington and Burkart, 2006) attached the unlabeled PP group and substrate to PCP1 enriched in $^{15}$N and $^{13}$C isotopes (Fig. 1). The NMR linewidths of the tethered moiety indicate that the arm does not tumble independently from the protein core but is also not rigidly docked onto

the protein, in line with a transient interaction. Unfortunately, we found that traditional filtering methods leave undesired labeled protein signals in the spectra of the PP arm and its substrate, confusing data interpretation and hampering future

mechanistic studies. Our immediate objective is to attenuate these residual signals and mitigate sensitivity losses for the targeted signals of unlabeled moieties, which will be particularly important for future studies of PCP1 engaging with its larger partner domains.

Isotopic filtering is a tested tool for detecting interactions between labeled and unlabeled molecules but is inefficient in

presence of large variations in scalar couplings. Many methodologies have been implemented to filter signals from labeled molecules in direct or indirect dimensions, reviewed in (Breeze, 2000; Robertson et al., 2009). As our immediate objective is to obtain 1D proton spectra of unlabeled moieties, we do not consider methods that exploit evolutions in indirect dimensions and here, we focus solely on filters for the detected dimension. The existing methods that achieve our aim rely on the same common principle: single-quantum coherences of protons coupled to heteronuclei can evolve into antiphase coherences

while those of protons in unlabeled molecules remain in-phase. The filtering pulse sequences are designed to preserve the latter and eliminate the former. This filtering methodology targets a specific range of scalar couplings and is limited when very different scalar couplings need to be targeted, as occurs for proton-carbon scalar couplings that vary from 120 to 220 Hz in proteins and 140 to 220 Hz in nucleic acids (Zwahlen et al., 1997). Three alternative solutions can overcome this challenge. Low-pass J filters (Kogler et al., 1983) can be incorporated (Ikura and Bax, 1992), elements targeting different

couplings can be applied sequentially (Breeze, 2000; Gemmecker et al., 1992; Zangger et al., 2003), and the correlation between chemical shift and scalar couplings can be exploited through adiabatic pulses (Eichmüller et al., 2001; Kupče and Freeman, 1997; Valentine et al., 2007; Zwahlen et al., 1997). In all cases, further suppression of labeled molecule signals comes at the cost of sensitivity losses, which we attempted to mitigate.

Here, we present a method to attenuate undesired signals in the detected dimension that escaped traditional filters with minimal increase in the length of the pulse sequence. The method relies on allowing evolutions under scalar couplings during both the detection period and existing adjacent preparation periods such as water suppression schemes. We show that, although only applicable to a narrow range of scalar couplings, this strategy satisfyingly removes spurious signals of the labeled protein core of PCP1 and provides reliable spectra of its tethered unlabeled phosphopantetheine group and substrate.

As an immediate application, we implemented our modification to a 2D-TOCSY experiment to remove misleading correlations. Due to the simplicity of its implementation, we predict that our method will readily improve studies of unlabeled moieties attached covalently or non-covalently to larger, labeled biomolecules.

## 2 Theory

### 2.1 Using delayed decoupling to suppress residual signals of protons coupled to heteronuclei

Our objective is to attenuate residual signals from coupled spins that have escaped filters with minimal or no increase in the lengths of pulse sequences. In our immediate application, we repurpose a water-suppression scheme in addition to the detection period to function as an additional filter. In this section, we discuss how signals are attenuated by sharing evolution under scalar couplings between these two periods and applying composite pulse decoupling once undesired coherence become antiphase.


Delayed decoupling can be used to dampen the signals of spins coupled to heteronuclei. Delayed decoupling has previously been used to enhance sensitivity in solution NMR (Rößler et al., 2020), and the partitioning of adjacent undecoupled and decoupled periods has been used to determine carbon hybridization states in solid-state NMR spectra (Alla and Lippmaa, 1976). Decoupling without delay has been used to suppress undesired antiphase coherences for filters immediately preceding

detection (Yang et al., 1995). Here, we use delayed decoupling to improve isotope filtering while minimizing sensitivity losses due to relaxation. Although our method is best implemented through experimental optimizations (*vide infra*), its mechanism can be described through simple principles. We first consider a simple pulse and acquire experiment, in which decoupling is applied after a time $\tau = 1/|2J|$ and assume that an initial in-phase single-quantum coherence has become fully antiphase by the end of $\tau$. Further, we only consider the signal of a single multiplet component. In absence of delayed

decoupling, the signal is simply:

$$s(t) = e^{(i\omega_0 - R)t} \quad , \tag{1}$$

where, for simplicity, $\omega_0$ implicitly includes contributions from the scalar coupling, i.e. for a two-spin system on resonance $\omega_0 = +/- \pi J$, and R is a transverse relaxation rate incorporating all relevant relaxation mechanisms. The signal in the frequency domain is that described in every NMR textbook:

$$S(\omega) = \frac{-1}{i(\omega_0 - \omega) - R}. \tag{2}$$

Note that we chose to use this compact form rather than the more conventional representation as a sum of real and imaginary components to remind the reader that the intensity at $\omega = \omega_0$ does not have any imaginary component, which will be exploited below and prevent unnecessary derivations. Thus, substituting $\omega$ with $\omega_0$ in Eq. (2) gives:

$$I(\omega = \omega_0) = \frac{1}{R}. \tag{3}$$

Again, a solution that should be familiar to every reader. When decoupling is applied after $\tau$, the signal only evolves during $\tau$, after which decoupling prevents evolution of antiphase coherences into detected in-phase terms. In our scheme, s(t) is zero after $\tau$ but has a conventional evolution during $\tau$. This description is reminiscent of discussions of truncation artefacts,

which, in the frequency domain, lead to the convolution of Lorentzian signals with a sinc function. That is, signals are now both attenuated and accompanied with sinc wiggles (Fig. 2(a)).

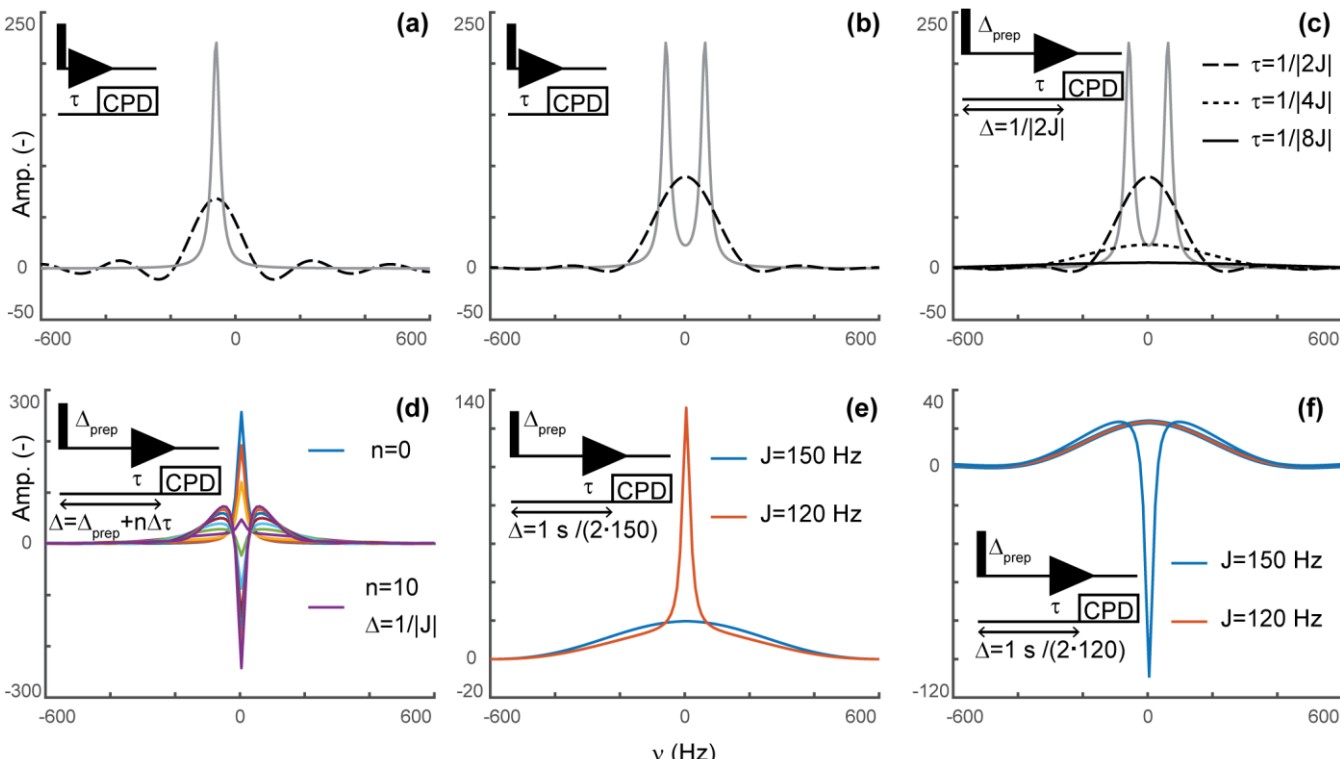


**Figure 2. Principles of editing through delayed decoupling.** (a) Applying decoupling once coherences are antiphase truncates their FID and attenuates their signals (dashed line), as shown here for the isolated component of a doublet. (b) The two components combine into a broadened and attenuated shape (dashed line). The analytical expressions of Eqs. (2) (solid grey line) and (4) (dashed black line) were used in (a) and (b). (c) Further attenuation is obtained when evolution into antiphase coherences is shared between a preparation period and detection as shown through simulations. The total evolution, $\Delta$, was set to $1/|2J|$, with evolutions during detection $\tau = 1/|2J|$ (dashed line), $1/|4J|$ (dotted line), and $1/|8J|$ (solid line). In (a)-(c), spectra without delayed decoupling are shown in grey for reference. (d) Simulation where the duration $\Delta$ is arrayed for a fixed preparation period $\Delta_{prep} = 1/|4J|$, and $\tau$ ranges from zero to $3/|4J|$ leading to $\Delta = 1/|J|$ in ten increments $\Delta\tau$ of $3/|40J|$. This simulation predicts the results seen in Fig. 4(b). In (a)-(d), J is set to 120 Hz. (e) A delayed decoupling targeting 150 Hz leads to residual positive in-phase signals for spins with couplings at 120 Hz. (f) A delayed decoupling targeting 120 Hz leads to negative residual in-phase signals for couplings at 150 Hz. In (e) and (f), $\Delta_{prep} = 1/|4J|$ and $\tau$ is set to $1/|4J|$ for the targeted J, i.e. half of the total duration $\Delta$.

An analytical expression can be obtained by integrating Eq. (1) between 0 and $\tau$ to give:

$$S_\tau(\omega) = \frac{e^{(i\omega_0 - R)\tau} - 1}{i(\omega_0 - \omega) - R},$$

(4)

providing the spectrum shown in Figure 2(a). The intensity at $\omega = \omega_0$ is

$$I_\tau(\omega = \omega_0) = \frac{1 - e^{-R\tau}}{R},$$

(5)

such that the delayed decoupling attenuates the intensity with

$$I_\tau(\omega = \omega_0)/I(\omega = \omega_0) = 1 - e^{-R\tau}.$$

(6)

That is, the efficiency of the attenuation is in part governed by relaxation and in part by the delay before decoupling is applied. For a system of two scalar-coupled spins, the signal corresponds to the sum of two shapes described by Eq. (5) and separated by J, as shown in Fig. 2(b). Although, in the example we discussed, $\tau$ is set to $1/|2J|$, our objective is to implement delayed decoupling in pulse sequences with preparation periods preceding detection. Equation (5) indicates that these preparation periods will enhance attenuations if undesired coherences are allowed to evolve under scalar couplings before detection such that smaller $\tau$ are needed to reach antiphase coherences.

Figure 2(c) depicts simulations describing how delayed decoupling affects signals of coherences when they also evolve under scalar couplings before detection. Here, we found the analytical solution to be of little use as the shape of the detected signal is a complex combination of multiplet components that are dephased during the preparation period, and the quality of the attenuation can no longer be assessed through the intensity at $\omega = \omega_0$. Instead, we ran simulations where the total duration from establishing a single-quantum coherence to applying decoupling is kept at $\Delta = \Delta_{prep} + \tau = 1/|2J|$, where $\Delta_{prep}$ is the evolution period preceding detection and $\tau$ is the delay described above, i.e. separating the beginning of signal detection from the application of decoupling. In Fig. 2(c), $\tau$ takes the values $1/|2J|$ (when $\Delta_{prep}$ is zero), $1/|4J|$, and $1/|8J|$. In agreement with Eq. (5), Fig. 2(c) indicates that signals are most attenuated when evolution towards antiphase occurs predominantly during the preparation period. However, it also shows that the shapes of the signals change dramatically, with a line broadening accompanied with attenuation of sinc wiggle artefacts.

Figure 2(d) shows a simulation where the duration between establishing the coherence and applying decoupling, $\Delta$, is arrayed for a target scalar coupling (here, 120 Hz). Figure 2(c) reports exclusively on sharing the evolution under scalar couplings between the preparation and detection periods but does not report on incomplete conversion into antiphase operators, as would occur when a variety of scalar couplings must be considered or when arraying delays to optimize signal attenuations, as performed experimentally. Figure 2(d) shows that when decoupling is applied before reaching $1/|2J|$, a residual positive in-phase signal is detected, whereas a negative in-phase signal emerges passed $1/|2J|$. These signals are combined to the shapes described above resulting from truncated antiphase evolution, leading to unconventional line-shapes, in particular, when $\Delta$ exceeds the optimal value of $1/|2J|$. Note that, because of this behaviour, zero amplitude is achieved on resonance for delays $\Delta$ slightly exceeding $1/|2J|$, when a residual negative in-phase signal compensates for the truncated antiphase signal (as seen for the signal in green in Fig. 2(d)). Indeed, experimentally, we found that the value of $\tau$ selected through visual inspection typically exceeds $1/|2J|$ as signal suppression appears more efficient at those values than at $1/|2J|$. Similarly, delayed decoupling filters tuned for a target scalar coupling will distort the shapes of signals with off-target couplings. This drawback is illustrated in Figs. 2(e) and 2(f) for filters targeting either the maximal *aliphatic* scalar coupling for [1]H-[13]C in proteins (around 150 Hz, on average) or the minimal value (120 Hz for methyls), respectively. As a corollary, the quality of the filter's bandwidth includes an aesthetic component which cannot reasonably be quantified. This aspect will

be illustrated experimentally below, with the major conclusion being that the filter is solely to be used to supplement existing filters and not replace them.

In conclusion to this section, delayed decoupling can dramatically reduce the signals of spins coupled to heteronuclei, but off-target couplings will display undesirable line-shapes. Thus, although not robust as a stand-alone technique, the method is ideal to supplement existing isotopic filters where it can target residual unwanted signals at reduced costs in sensitivity for the signals of unlabeled moieties.

## 3 Methods

### 3.1 Simulations

Simulations were coded in Matlab (Matlab Inc., 2018). Briefly, the evolution of the density operator is calculated in a single-transition and polarization operator basis, only accounting for the detected terms $I.S^{\alpha}$ and $I.S^{\beta}$. Propagation is achieved through the matrix:

$$M = \begin{pmatrix} i\omega_{0\alpha} + R & 0 \\ 0 & i\omega_{0\beta} + R \end{pmatrix}.$$

For simplicity, the simulation is only performed on resonance such that $\omega_{0\alpha} = \pi J$ and $\omega_{0\beta} = -\pi J$. R is a transverse relaxation rate, either obtained through calculations or estimated experimentally. In our simulations, we used a value of $2\pi \cdot 14$ Hz estimated from the linewidth of an isolated signal. The density operator at the time of detection was calculated through exp(-$M\Delta_{prep}$)$S_0$, where $S_0$ was set to (1;1). With the simplifications described above, this propagation describes the evolution into antiphase operators with concomitant refocusing of chemical shift evolutions. An array recording the evolution during the detection period is then calculated through exp(-Mdt)S(t-dt), first with free evolution, and from the time t = $\tau$, with intermittent exchange of $I.S^{\alpha}$ and $I.S^{\beta}$, thus simulating decoupling through a perfect 180° pulse. The relationship between the dwell time and dt are set by the number of inversions during the dwell time, and in the present work the dwell is 2dt for one inversion per dwell. Departure from the simulations include cross-correlated autorelaxation and autocorrelated cross-relaxation effects that would differentiate the relaxation rates of each component and mix the two components, respectively. Further differences with experimental implementations include evolutions during composite pulse decoupling (CPD) sequences and off-resonance effects.

### 3.2 Sample Preparation

### 3.2.1 PCP1 cloning, isotope labelling, expression, and purification

The PCP1 construct used in this study was prepared as described in (Harden and Frueh, 2017) utilizing the 1402-1482 gene fragment from the Yersinia pestis irp2 gene (Accession Number AAM85957). Briefly, PCP1 (9.6 kDa) is expressed as a

His$_6$-GB1 fusion protein containing a Tobacco Etch Virus (TEV) cleavage site. Following expression of the protein in an E. coli BL21 (DE3) competent cell line (courtesy of Drs. Chalut and Guilhot, CNRS, Toulouse, France) in M9 minimal media containing 1 g/L $^{15}$NH$_4$Cl and 2 g/L $^{13}$C glucose, the protein is lysed, purified through His-affinity, cleaved with TEV protease to remove the GB1 tag, and purified by size-exclusion chromatography in phosphate buffer (20 mM sodium phosphate, pH 6.60 at 22 °C, 150 mM NaCl, 1 mM DTT, and 1 mM EDTA). The protein was flash-frozen in liquid nitrogen and stored at -80 °C before use in the one-pot loading protocol.

### 3.2.2 One-pot chemoezymatic synthesis of Cys-loaded PCP1

One-pot loading of apo PCP1 ($^{15}$N/$^{13}$C) with the cysteine-linked pantetheine analogue (courtesy of Drs. David Meyers and Yousang Hwang, Johns Hopkins chemistry core facility) followed methods described by (Worthington and Burkart, 2006) and improved by (Kittilä and Cryle, 2018) with the following adaptations. Apo PCP1 was thawed and buffer exchanged into one-pot reaction buffer: 100 mM Tris pH 7.55 at 22 °C, 10 mM MgCl$_2$, 100 mM NaCl, and 2.5 mM DTT. To a 10-mL reaction at 25 °C, apo PCP1 (118 μM), Cys-NH-pantetheine analogue (509.4 μM), ATP (1 mM), PanK (2.5 μM), PPAT (2.5 μM), DPCK (2.5 μM), and Sfp R4-4 (5.0 μM) were mixed together following sequential addition-incubation steps as in (Kittilä and Cryle, 2018). Preparation of PanK, PPAT, and DPCK was performed as described (Goodrich et al., 2017). Sfp R4-4 (courtesy of Dr. Jun Yin, Georgia State University) was prepared as described in (Yin et al., 2006). The reaction incubated in a water bath at 25 °C for two hours. Alkaline phosphatase (Calf Intestine, Quick CIP, New England BioLabs) was added to 1.0 U/mL and the reaction was monitored by NMR until loading completion was observed. Following this incubation period, 2D HN-HSQC NMR and MALDI-TOF mass spectra were used to determine loading efficiency (Fig. A1 and A2). Trace amounts of apo PCP1 are estimated to be less than 1% (Fig. A2). The reaction mixture was then loaded onto a 5-mL HisTrap column (GE Healthcare) to remove PanK, PPAT, and Sfp R4-4, all of which have a His$_6$ affinity tag. The eluted PCP1 was concentrated and purified by size-exclusion chromatography on a Superdex 75 16/60 pg column (GE Healthcare) equilibrated with one-pot reaction buffer. Upon confirmation of loading, purified Cys-loaded PCP1 (10 kDa with the prosthetic group) was concentrated and buffer exchanged into NMR buffer containing 20 mM sodium phosphate pH 6.59 at 22 °C, 150 mM NaCl, 1 mM EDTA and 2 mM TCEP. All NMR experiments were performed on a 314 μM sample of PCP1 containing 10% D$_2$O and 200 μM DSS. PCP1 concentration was quantified using UV-Visible absorbance at 280 nm using an extinction coefficient of 6990 M$^{-1}$ cm$^{-1}$.

### 3.3 Data acquisition

All NMR experiments were performed at 25 °C on a 600 MHz Bruker Avance III spectrometer equipped with a QCI cryoprobe using a 314 μM sample of PCP1 ($^{15}$N/$^{13}$C) covalently modified with a non-hydrolyzable analogue of cysteine-linked phosphopantetheine (see Sect. 3.2). All 1D and 2D experiments were collected with 128 transients using 3072 points during detection. All 1D spectra were zero-filled to 4096 points before Fourier transform and subsequently apodized using exponential multiplication with 7 Hz broadening to reduce truncation artefacts from buffer signals. In all experiments, water

suppression was performed with a 3-9-19 WATERGATE element (Piotto et al., 1992) (see Fig. 3). Further details of water suppression and pulse/delay parameters can be found in Sect. 4.1. All 1D spectra were processed in TopSpin 4.0.7 (Bruker BioSpin, 2019).

Implementation of the $X_{J1}X_{J2}X_{d,J3}$ combined filter into the 2D $^1$H-$^1$H TOCSY was done by incorporating our scheme into the WATERGATE sequence in a 2D homonuclear X-filtered TOCSY pulse sequence (modifying the Bruker code dipsi2gpphwgxf) (Breeze, 2000; Iwahara et al., 2001; Ogura et al., 1996; Zwahlen et al., 1997). This pulse sequence includes t1-encoding, a TOCSY mixing time using a DIPSI-2 sequence (Rucker and Shaka, 1989), followed by a sequential, double-

tuned X-filter and t2-encoding. A second version was coded to incorporate the $X_d$ block using an optimized $\tau$ of 431 μs. Both 2D TOCSY spectra were collected with 1536 t1 and 128 t2 complex points using a spectral width of 16.0192 ppm in the detected dimension and 10.012 ppm in the indirect dimension. Quadrature detection in the indirect TOCSY dimension was achieved using States-TPPI (Marion et al., 1989). The field strength of the DIPSI-2 TOCSY mixing sequence was 10.008 kHz and the mixing time was set to 40 ms. Both 2D spectra were processed by zero-filling to 4096 points in the detected

dimension and 1024 points in the indirect dimension. Before Fourier transform, spectra were apodized with a cosine-squared bell window function. The detected dimension was solvent corrected using polynomial subtraction and extracted over the region from 9.0 to 6.5 ppm for focus on aromatic signals. All data were processed in NMRpipe (Delaglio et al., 1995) and referenced in both proton dimensions using the frequency of DSS.

## 4 Results and Discussion

### 4.1 description of pulse sequence elements

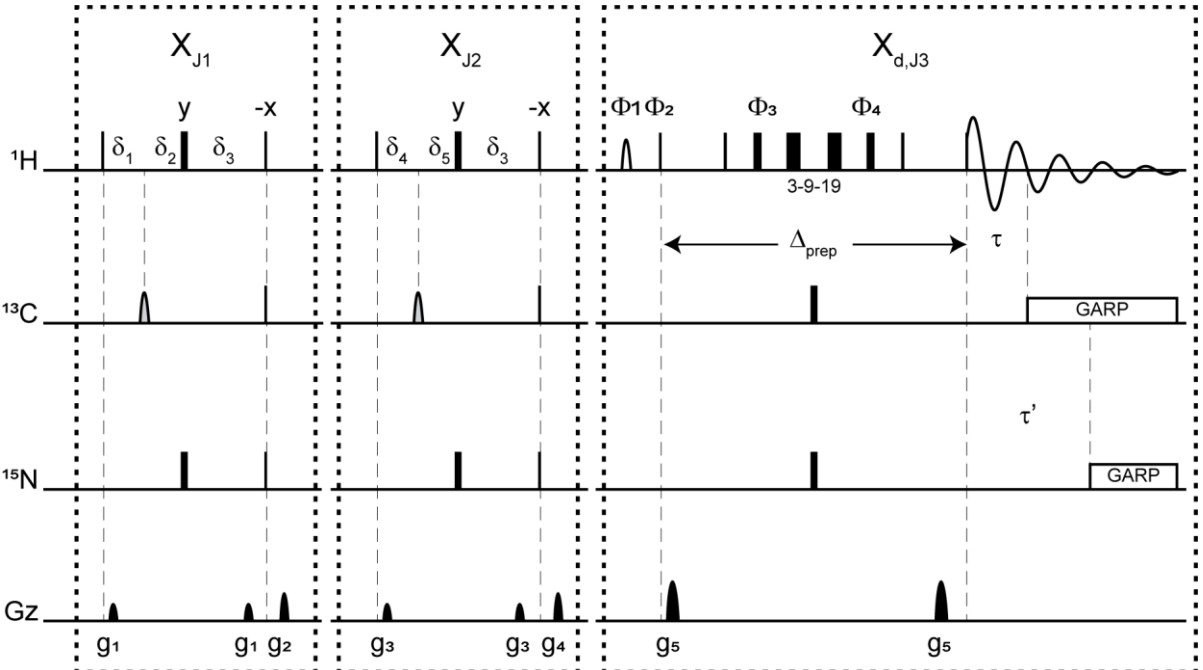

**Figure 3.** Pulse sequence for a 1D double X-half-filter with delayed decoupling for suppression of $^{13}$C (and/or $^{15}$N) labeled protein signals and preservation of unlabeled moieties signals. Dashes indicate different filtering blocks that can be used modularly to suppress scalar coupled spin systems. Here, X denotes isotopic half-filters and the sub-scripts J1, J2, and J3 indicate individually targeted scalar couplings. The subscript d indicates our delayed decoupling method. Thin and thick rectangles indicate 90° and 180° hard pulses, respectively. Unless otherwise noted, the phase of these pulses is along the *x* axis. Open, half ellipses pulses on carbon are 180° frequency swept chirp inversion pulses (Böhlen and Bodenhausen, 1993), with durations of either 500 μs, when using sequential tuned filters, or 1730 μs, when using the chemical shift-coupling matched sweeping rate (Zwahlen et al., 1997). In Bruker, the shaped pulses were Crp60,0.5,20.1 and Crp60_xfilt.2, respectively. Water suppression is achieved through a WATERGATE element in the $X_{d,J3}$ filter employing a 3-9-19 suppression scheme (Piotto et al., 1992). A water flipped-back pulse (Grzesiek and Bax, 1993) (denoted with a curved open half-ellipse) is applied on resonance with water (4.699 ppm) using a 1.5 ms 90° Sinc pulse. The delays in the $X_{J1}$ and $X_{J2}$ filter blocks are: $\delta_3 = 1/|4\ J(NH)|$ $\approx 2.78$ ms, $\delta_1 = 1/|4\ J1(CH)|$, $\delta_4 = 1/|4\ J2(CH)|$, $\delta_2 = 1/|4\ J(NH)| - 1/|4\ J1(CH)|$, and $\delta_5 = 1/|4\ J(NH)| - 1/|4\ J2(CH)|$. The delays $\tau$ and $\tau'$ in the $X_{d,J3}$ delayed decoupling block can each be manually set to target residual $^1$H-$^{13}$C and $^1$H-$^{15}$N signals, respectively. Although we did not need this feature, residual $^1$H-$^{15}$N signals can be suppressed using a shared delayed decoupling element that is included in our code. When suppression of surviving $^1$H-$^{15}$N signals is not needed, both the 180° pulse and decoupling sequence on the nitrogen channel can be switched off. The two delays $\tau$ and $\tau'$ are arrayed to find the optimal level of suppression of targeted signals. Decoupling on carbon is performed by applying a GARP pulse train (Shaka et al., 1985) on resonance with the targeted signals using a bandwidth of 2.083 kHz. Nitrogen decoupling is applied through a GARP sequence using a bandwidth of 1.042 kHz. The phase cycling in the delayed decoupling element is: $\Phi_1$ = -x, x, $\Phi_2$ = x, -x,  $\Phi_3$ = x, x, y, y, -x, -x, -y, -y, $\Phi_4$ = -x, -x, -y, -y, x, x, y, y,  and $\Phi_{REC}$ = x, -x, -x, x. Gradients are applied with lengths ($\tau g_i$) and power ($g_i$) of: $\tau g_1 = \tau g_3 = \tau g_5 = 300$ μs (Bruker shape SMSQ10.32), $\tau g_2 = \tau g_4 = 1$ ms (Bruker shape SMSQ10.100), $g_1 = 2.5$ G/cm, $g_2 = 5.5$ G/cm, $g_3 = 1.665$ G/cm, $g_4 = 3.5$ G/cm, and $g_5 = 30$ G/cm. All gradients are followed with a 200 μs gradient recovery delay.

Figure 3 displays the pulse sequence elements used to assess the improvement provided by our method and, in the end, provide spectra of our unlabeled moiety free from signals of the labeled protein core. The novel element is labeled $X_{d,\ J3}$, where d emphasizes that the filter ends during the detection, and J3 denotes the scalar coupling it targets. By analogy, we

label each traditional X half-filter as $X_{J1}$ and $X_{J2}$, where J1 and J2 denote the couplings they target. When the Zwahlen method is used (Zwahlen et al., 1997), we label these periods $X_Z$. In reference experiments, the $X_d$ block is replaced by a 3-9-19 water suppression scheme, thus keeping all pulse sequences the same length for comparison. This consideration ensures that attenuations in signal intensities report exclusively on the efficiency of the filter and not on relaxation. The 3-9-19 scheme simply omits the inversion pulses on $^{13}C$ and $^{15}N$ shown in the $X_{d,J3}$ block, as well as the delayed composite pulse decoupling sequences.

Each of the $X_{J1}$ and $X_{J2}$ elements is an updated X half-filter. When applied sequentially, they provide an update of the original sequential tuned filter (Gemmecker et al., 1992) as described in (Breeze, 2000). Briefly, while single-quantum coherences in the isotopically labeled system evolve under scalar couplings, those in the unlabeled molecule will not. Thus, at the end of the delay $\delta 3$, protons in labeled molecules feature antiphase coherences orthogonal to in-phase coherences of protons of unlabeled molecules. The desired in-phase coherences are then converted into longitudinal magnetization while a pulsed-field gradient dephases the coherences of labeled molecules, now in multiple-quantum coherences. Variations in scalar couplings across the targeted molecule lead to imperfect signal suppressions, as occurs for protons coupled to $^{13}C$ in proteins, and several strategies have been developed to overcome this challenge (Breeze, 2000; Robertson et al., 2009). Two such strategies will form the basis for our comparisons.

When both blocks are applied sequentially with different values for J1 and J2, we obtain an updated version of the original sequential tuned filter of Gemmecker et al. (Gemmecker, Olejniczak, and Fesik 1992). The principle is simple: multiple elements, each tuned to a target scalar coupling, are applied sequentially to cover a wider range of scalar couplings. With enough blocks, and with variations in tuned delays for each transient, remarkable editing can be achieved (Zangger et al., 2003) at the cost, however, of sensitivity losses due to relaxation during spin manipulations. In this study, to minimize such losses, only two blocks are considered. Our final implementation can be regarded as including a third block that is shared between the last preparation period and the detection period.

When using the chemical shift-coupling matched sweeping rate in chirp pulses, we obtain a variant of the pulse sequence of Zwahlen et al. (Zwahlen et al., 1997). Here, the values of J1 and J2 are both set to a value of 147.4 Hz under our conditions, corresponding to $\tau a = 1.696$ ms in the original paper. The main difference with the original paper is that the length of each block in our pulse sequence is maintained to $1/|2J(NH)|$ so as to permit comparisons with the sequential tuned filter without interferences from relaxation losses. In either solution, time-shared filtering of amide protons is implemented (Burgering et al., 1993; Ikura and Bax, 1992).

The $X_d$ block implements our method into water suppression schemes. Incorporation of water suppression schemes in X-half filters has already been described (Breeze, 2000; Sattler et al., 1999). Briefly, inversion pulses are applied on $^{13}C$ and $^{15}N$

concomitantly with the existing proton inversion, here in the form of a 3-9-19 sequence, to enable evolution under scalar couplings. In our strategy, composite pulse decoupling is then delayed until coherences have become antiphase during detection. The outcome of this strategy is to reduce the length of a filter preceding the detection period by sharing the evolution into antiphase coherences with the detection period, thus mitigating relaxation losses. For measurements in $D_2O$, X-half-filters preceding the detection period can be implemented as $X_d$ blocks to reduce their lengths.

## 4.2 Experimental implementation of delayed decoupling

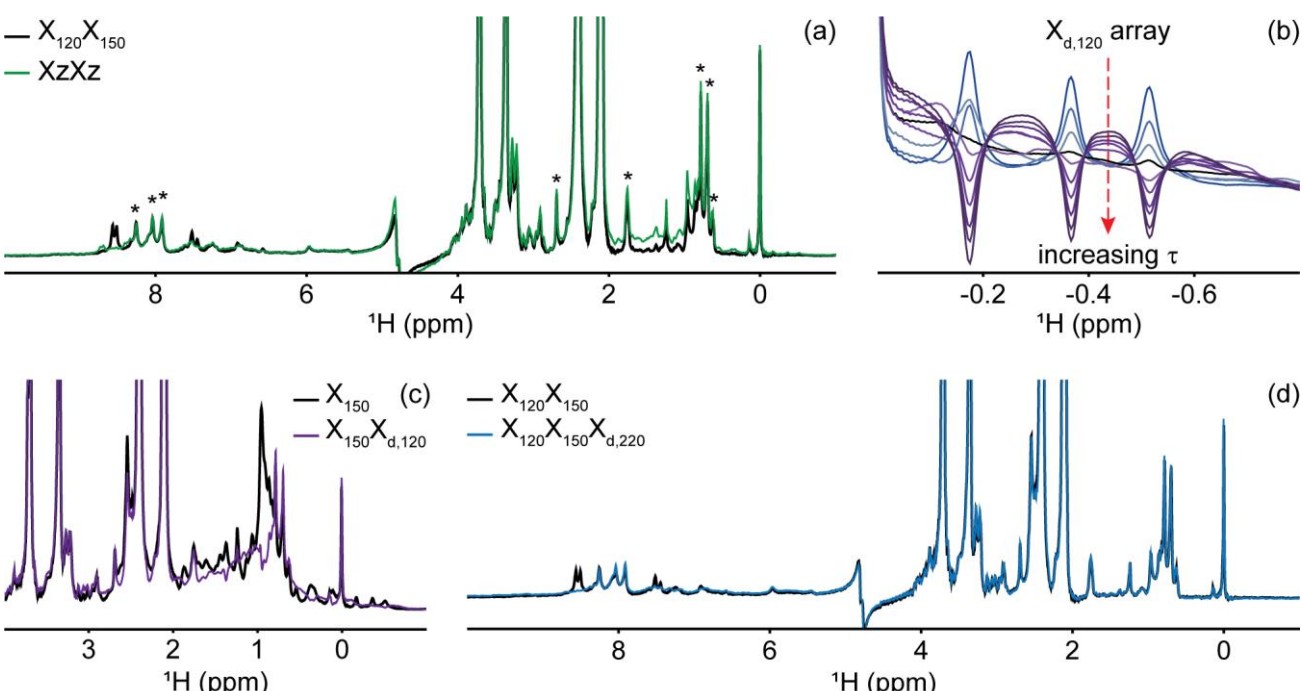

**Figure 4. Delayed decoupling removes residual protein signals escaping traditional filters.** (a) We compared suppression of aliphatic and aromatic signals from the core of PCP1 ($^{15}N$ and $^{13}C$ labeled) by using a frequency-swept adiabatic pulse as referenced in (Zwahlen et
al., 1997) versus the level of suppression when employing a sequential double X-half-filter targeting 120 and 150 Hz in the aliphatic region. (b) Our delayed decoupling element can be arrayed to target specific scalar-coupled spins, here shown for PCP1 methyl peaks. $\tau$ ($\mu s$): 3 (blue), 1340, 1722, 2295, 2486, 3250, 3632, 4205, 4587, and 5542 (purple). $\Delta_{prep}$ is set to 2.3939 ms. Optimal suppression is achieved for $\tau$ set to 2.295 ms corresponding to a total evolution time of 4.6889 ms (black). (c) Combining an $X_{150}$ block with an $X_{d,120}$ block (using the optimal value determined in (b)) supresses methyl resonances but underperforms the $X_{150}X_{120}$ scheme shown in (a) for
scalar couplings departing from the targeted value. (d) Final $X_{120}X_{150}X_{d,220}$ filter with optimal suppression in both aromatic and aliphatic regions. $\tau$ was set to 329 $\mu s$, corresponding to a total evolution of 2.7229 ms. Peaks marked with asterisks in panel (a) indicate signals from the prosthetic group that are left untouched by the targeted filter elements. Other signals of unlabeled molecules belong to buffer components (EDTA and TCEP, with larger intensities) and DSS (at 0 ppm). Implementation of $X_{J1}$, $X_{J1}X_{J2}$, $X_ZX_Z$, and $X_{J1}X_{J2}X_{d,J3}$ filters were performed as described in the pulse sequence (Fig. 3). The pulse sequences used to obtain all spectra that are compared have the
same lengths, and the comparisons report exclusively on the efficiency of the filters.

A large range of $^1J_{CH}$ scalar couplings hampers optimal suppression of coherences through traditional X-half-filter elements.

We first set to compare the performances of established strategies for our system. We did not test low-pass filters as Zwahlen

at al. (Zwahlen et al., 1997) already demonstrated that their method outperforms it. Similarly, we did not implement improved sequential tuned filters (Zangger et al., 2003) as we recognized that our method could provide a means to include the third filter needed for this improvement without an increase in pulse sequence length, and hence both methods should rather be combined than compared. Our two reference experiments are a double sequential tuned half-filter, $X_{150}X_{120}$, targeting 150 Hz and 120 Hz scalar couplings, and the Zwahlen experiment $X_ZX_Z$ (see Sect. 4.1 for departures from published sequences). Figure 4(a) reveals that each method is subject to orthogonal drawbacks. Globally, the Zwahlen method outperforms the sequential filter as both aliphatic and aromatic regions are filtered. However, any chemical moiety with a deviation from the correlation between the scalar coupling and chemical shift used to optimize the sweep rate will display a residual signal, as observed most prominently in the aliphatic region. We note that we did not implement alternative means to exploit the correlation between scalar couplings and chemical shifts (Eichmüller et al., 2001; Kupče and Freeman, 1997; Stuart et al., 1999; Valentine et al., 2007), as outlier residues would still escape the filters, regardless of improvements in how the sweep is achieved. Figure 4(a) shows that using two sequential X-filters tuned for 120 Hz and 150 Hz outperforms the Zwahlen method in the aliphatic region. However, this advantage is offset by a near complete lack of suppression for aromatic moieties (Fig. 4(a), 6 to 9 ppm). Our observations reinforce previous comparisons (Zangger et al., 2003) but recording the references were necessary to identify signals escaping filters in our systems. In the remainder of this section, we will implement our $X_d$ block to a sequential tuned filter to benefit from its superior suppression in the aliphatic region, whilst using $X_d$ to compensate for its deficiency in the aromatic region. Below, we implement $X_d$ gradually to combinations of $X_{J1}$ and $X_{J2}$ blocks to illustrate its performance and limitations, beginning with testing $X_d$ on its own.

Delayed decoupling is a tool to be combined with traditional X-filter elements to eliminate residual signals of labeled molecules. To test our $X_{d,J3}$ scheme and verify our theoretical predictions, we first incorporated it into a simple WATERGATE scheme (Piotto et al., 1992) and focused on isolated methyl resonances from PCP1 (Fig. 4(b)). This corresponds to a single X-filter, including water suppression (Breeze, 2000; Sattler et al., 1999), taking place during both the preparation and detection period. Here, we experimentally arrayed the delay for delayed decoupling, $\tau$, to emulate our simulations (Fig. 2(d)) and observed near complete suppression (black in Fig. 4(b)) when $\tau$ slightly exceeds the value calculated from the scalar coupling estimated from a non-decoupled spectrum (122 Hz), as discussed in Sect. 2. The differences between the line shapes of Figs. 2(d) and 4(b) reflect signal overlap and apodization. That all three resonances are suppressed at the same delay $\tau$ reflects how close their scalar couplings are. To illustrate the experimental limitations of our $X_d$ block, we paired it with a single $X_{150}$ block to make an $X_{150}X_{d,120}$ sequence (Fig. 4(c)). Comparing the results with that of Fig. 4(a) for $X_{150}X_{120}$ reports on the trade-off for including evolution into antiphase coherences during the detection period. In agreement with Sect. 2, $X_{150}X_{d,120}$ achieves adequate suppression for the targeted scalar coupling (three isolated resonances also shown in Fig. 4(b)) but severely underperforms a conventional $X_{150}X_{120}$ otherwise. Thus, although it serves its purpose to eliminate signals associated with specific scalar couplings, the method is not to be used as an alternative to existing methods.

Inclusion of delayed decoupling in the detected dimension of a sequential tuned filter removes spurious signals with minimal costs in sensitivity and improves the quality of related multi-dimensional experiments. As revealed in Fig. 4(a), the

360 sequential $X_{150}X_{120}$ sequence adequately supresses signals in the aliphatic region but signals of aromatic protons escape filtering. Thus, we incorporated our $X_{d,J3}$ scheme to that sequence and coded an $X_{150}X_{120}X_{d,220}$ sequence. Effectively, this sequence can be regarded as a variation of the triple-tuned filters of (Zangger et al., 2003), in which the last filter is shared between the water suppression scheme and the detection period. Here, $\tau$ was arrayed and optimal suppression was achieved for a value corresponding to 183.6 Hz. Figure 4(d) shows that $X_d$ could suppress surviving signals from the protein by up to

365 83% in the aromatic region (calculated from intensities) and the sequence will permit unambiguous studies of the unlabeled tethered prosthetic group. As an immediate application, we show enhanced suppression of undesired PCP1 signals when $X_d$ is incorporated into the direct dimension of a sequential X-half-filtered 2D TOCSY experiment (Fig. 5). To demonstrate the improvement, the 2D TOCSY was first run using a conventional $X_{120}X_{150}$ filter element (Fig. 5, black) and compared with a 2D TOCSY using an $X_{120}X_{150}X_{d,220}$ element (Fig. 5, blue) that additionally targets residual signals in the aromatic region. A

370 simple visual inspection reveals that spurious cross peaks between protein aromatic signals are efficiently suppressed upon addition of the delayed decoupling element, while cross peaks observed between the amide protons in the phosphopantetheine arm and aliphatic neighbours (around 3.5 ppm along the Y axis) do not suffer from losses in sensitivity. Thus, in spite of its limitation and perhaps esoteric nature, we find delayed decoupling as a filter to be straightforward to implement and to immediately improve on existing filtering techniques aimed at studies of unlabeled moieties in presence of

375 labeled partners.

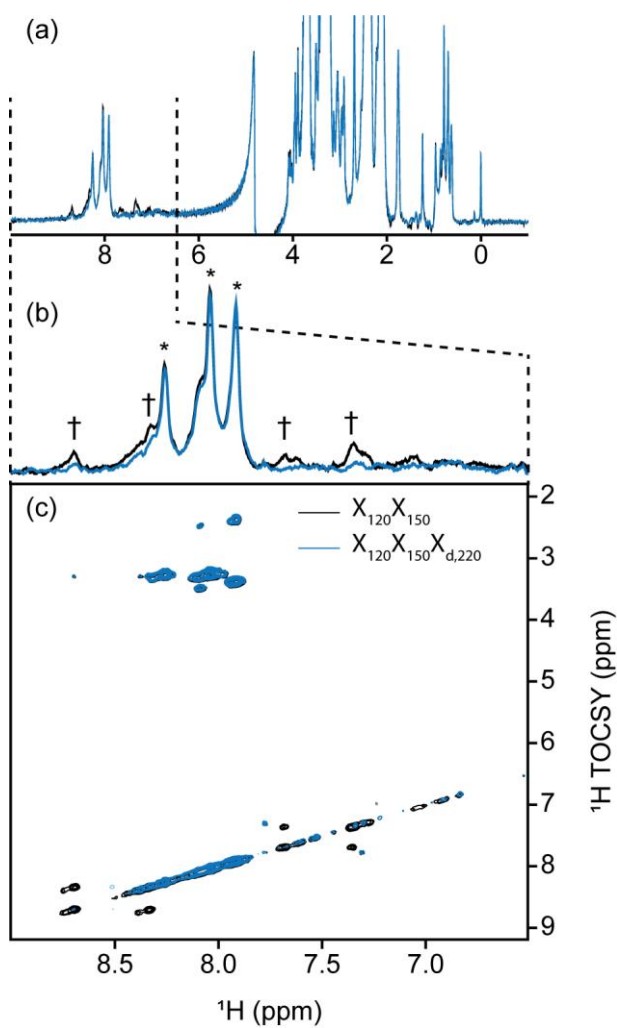

**Figure 5. Implementation of delayed decoupling into 2D homonuclear TOCSY.** $X_{150}X_{120}$-TOCSY were recorded with (blue) and without (black) a $X_{d,220}$ filter for enhanced suppression of aromatic signals. (a) 1D trace of the detected dimension. The aliphatic region is identical in both spectra but $X_d$ improves the suppression of aromatic signals. (b) Zoom of the spectral region from 6.5 to 9.0 ppm emphasizing the reduction of aromatic signals. Daggers denote aromatic signals from PCP1 that are not suppressed in the TOCSY-$X_{120}X_{150}$ experiment lacking $X_d$. Asterisks denote signals from the prosthetic that are preserved. (c) Cross peaks between aromatic signals of PCP1 are largely suppressed in the 2D-TOCSY upon addition of our delayed decoupling technique with no loss in signal sensitivity for unlabeled signals.

## 5 Future Directions

Delayed decoupling can readily be incorporated into existing water suppression schemes, and hence implemented into traditional multidimensional experiments, e.g. NOESY or COSY, with detection of unlabeled moieties. Notably, X-filtering techniques are routinely employed with NOESY experiments to provide correlations between labeled and unlabeled moieties (van Ingen et al., 2002; Karimi-Nejad et al., 1999; Ogura et al., 1996; Otting and Wüthrich, 1990; Petros et al., 1992) and

reveal binding sites or permit structure determinations of complexes. More generally, we anticipate this method will readily improve studies that focus on molecular responses of unlabeled moieties in presence of labeled partners. Ligand binding studies should benefit from this advance, in particular for tight binding when alternatives based on translational and rotational diffusion will fail. We will make immediate use of this filtering methodology to monitor chemical modifications of carrier proteins, e.g. to monitor the progress of the reaction described in Fig. 1. Notably, although well established (Kittilä and Cryle, 2018; Worthington and Burkart, 2006), the method requires a series of active enzymes and any defective component delays or abrogates the global reaction. With our improved scheme, we will be able to monitor steps of this reaction without interferences from residual protein signals. Further experiments using these improved filters will enable studies of interactions between the prosthetic arm and PCP1, in isolation and in the presence of its catalytic partner domains.

## 6 Conclusion

We have demonstrated that combining delayed decoupling with existing X-half-filters improves the suppression of labeled signals while preserving unlabeled signals in mixed samples. We have shown that the delayed decoupling technique can be easily shared between WATERGATE elements that are routinely used to study proteins by solution NMR and the detection period. Although only efficient over a narrow range of scalar couplings, and hence of little use as a stand-alone method, the method is complementary to existing filters. Specific scalar couplings that survive pre-existing X-filters are optimally and easily suppressed by arraying a delay in the pulse sequence. We anticipate that our technique will facilitate studies of post-translational modifications or protein/small molecule interactions and will help monitor *in situ* chemical reactions targeting macromolecules.

# 7 Appendices

## 7.1 Appendix A

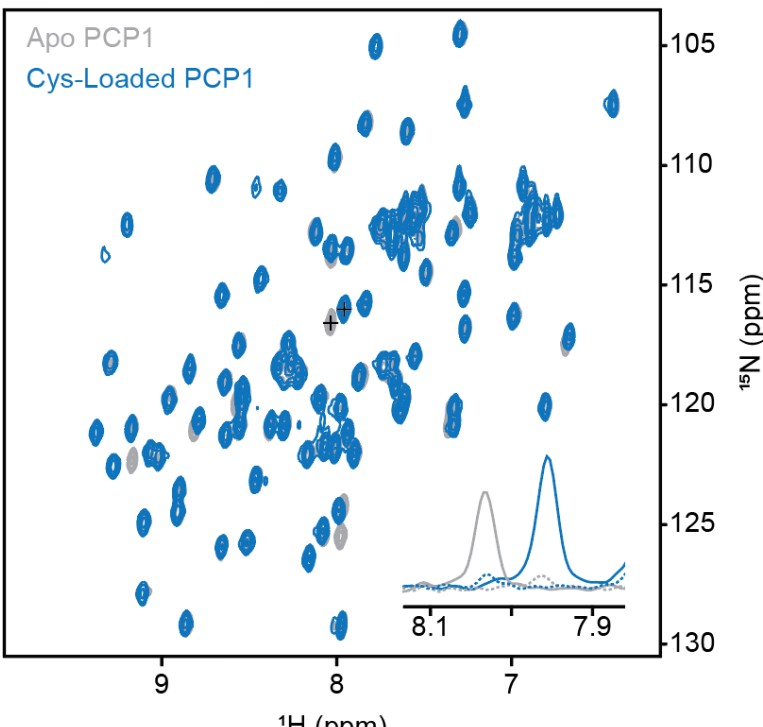

**Figure A1. Conformation of loading of PCP1.** Loading of PCP1 ($^{15}$N/$^{13}$C) with the phosphopantetheine analogue was confirmed through 2D HN-HSQC NMR. Upon loading of PCP1, the spectrum of apo PCP1 (grey) shows several characteristic peak shifts that can be used to monitor the chemoenzymatic loading of the prosthetic group (blue) detailed in Sect. 3.2. The inset shows 1D slices of a PCP1 residue that reports on loading (peaks marked with crosses). Solid lines indicate slices at signal maxima and dashed lines indicate slices at the same position when signals are weak or non-existent.

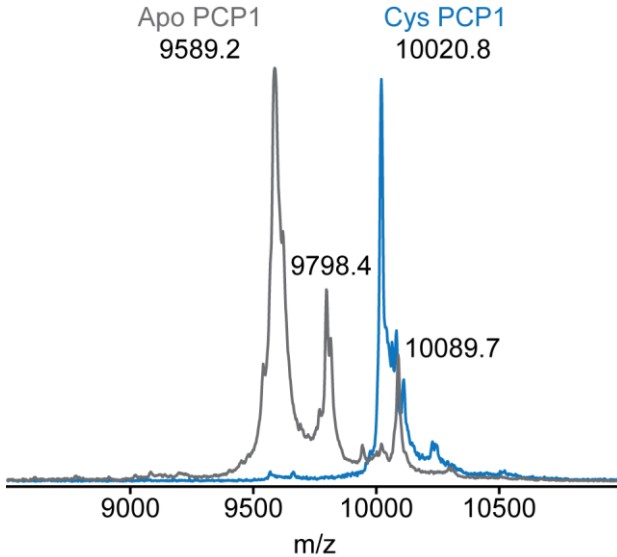


**Figure A2. MALDI-TOF Mass spectrometry is used to confirm loading of PCP1.** To confirm loading of PCP1 with an orthogonal method, the mass-to-charge ratio of PCP1 approximates loading of the prosthetic group. Apo PCP1 shows a characteristic mass shift upon loading of the prosthetic. Here, an experimental mass change of 431.6 Da was observed. A theoretical mass change of 426.4 Da is expected upon loading of PCP1. Peaks at a m/z of 9798.4 and 10089.7 in apo are artefacts of MALDI sample preparation (e.g. matrix

adduct with sinapinic acid). For comparison, the MALDI-TOF spectrum of Cys PCP1 has been scaled up by 3 times as the sample was more dilute. All MALDI-TOF spectra were collected on a Voyager DE-STR (Applied Biosystems) spectrometer in linear mode using a mass range of 2500 to 25000 Da. Sinapinic acid (10 mg/mL in 50% acetonitrile/0.1% trifluoroacetic acid (TFA)) was applied as sample matrix using a 10-fold dilution of protein sample in 0.1% TFA. Spectra were calibrated against standards ranging from 5734.51 to 16952.30 Da (Protein Calibration Standard I, Bruker).


**Code and Data Availability.** All NMR data including acquisition parameters, pulse sequences, and NMRPipe processing scripts, Matlab codes, and MALDI-TOF data are available for download from Zenodo at http://doi.org/10.5281/zenodo.4730836 (Marincin et al., 2021).

**Author Contributions.** DF, KM and IP designed experiments, and KM and IP prepared NMR samples. KM recorded the
final MALDI-TOF MS and NMR experiments and analysed the data. DF developed the theory and carried out simulations. KM and DF wrote the manuscript with feedback from IP. All authors read and approved the final manuscript.

**Competing Interests.** The authors declare that they have no conflict of interest.

**Acknowledgements.** We thank Drs. David Meyers and Yousang Hwang for preparation of the pantetheine analogue and Dr. Wolfgang Bermel for assistance in pulse program coding of delayed decoupling.

**Financial Support.** The research provided in this publication was supported by the National Institute of General Medical Sciences of the National Institutes of Health under award number R01GM104257.

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
