# Peer review of "Using delayed decoupling to attenuate residual signals in editing filters"

_Magnetic Resonance, 2021_

## Author Response (AR1)

*Referee responses are denoted in bold text.*

Referee #1:

**"Isotope filtration is a widely used technique in NMR spectroscopy to suppress signals from 1H spins attached to 13C or 15N spins. The variation particularly in 1H-13C scalar coupling constants poses a challenge to isotope filtrantion methods in solution NMR spectroscopy and unfortunately techniques to compensate for variations in couplings are typically less sensitive than simpler methods (better compensation results in longer pulse sequences). The present paper uses delayed decoupling to provide an additional level of suppression in a pulse sequence implementing standard filtration methods. The method is well-described and validated by the authors and promises to be very valuable in the study of complexes between isotopically enriched and unlabeled species. My only comments on this well written paper would be to suggest that references be added for**

**Madis Alla, Endel Lippmaa, High resolution broad line 13C NMR and relaxation in solid norbornadiene, Chemical Physics Letters, 37, 260-264, 1976.**

**which introduces delayed decoupling for spectral editing in CPMAS solid state NMR and**

**Rößler P, Mathieu D, Gossert AD. Enabling NMR Studies of High Molecular Weight Systems Without the Need for Deuteration: The XL-ALSOFAST Experiment with Delayed Decoupling. Angew Chem Int Ed Engl. 59, 9329-19337, 2020.**

**which uses delayed decoupling to enhance sensitivity in solution heterocorrelation experiments."**

Response:

We thank the referee for a timely and very supportive review. We agree that delayed decoupling is uncommon, and we will cite other applications when we first mention the term "delayed decoupling":

"Delayed decoupling has previously been used to enhance sensitivity in solution NMR (Rößler et al., 2020), and the partitioning of adjacent undecoupled and decoupled periods has been used to determine carbon hybridization states in solid-state NMR spectra (Alla and Lippmaa, 1976). Decoupling without delay has been used to suppress undesired antiphase coherences for filters immediately preceding detection (Yang et al., 1995). Here, we use delayed decoupling to improve isotope filtering while minimizing sensitivity losses due to relaxation."

Referee #2:

We thank the referee for the thorough and thoughtful review of our manuscript. We agree with all points raised and made modifications and clarifications to address these concerns. Please find a response to every point raised below.

**1. Regarding the model system (PCP1:pantatheinate covalent adduct) it would be useful if they could confirm whether the adduct interacts with the protein (i.e. tumbles at the macromolecular rate), or whether it is mobile relative to the protein (the latter might present an easier case for isotope-filtering because of the reduced relaxation penalty incurred by extra filter delays). Also, what is the size of the complex (about 20kDa?)**

Response: This is a good point, as we previously observed transient interactions between the prosthetic group and a different carrier protein (an aryl carrier protein). That is, the phosphopantetheine (PP) and its attached substrate sample both an undocked state and a docked state. However, we have not yet quantified this interaction equilibrium for PCP1. At this stage, we can only observe that the signals of PP and its substrate are broadened upon attachment, which may reflect a predominantly docked form or exchange line-broadening due to the docked/undocked equilibrium or both. Importantly, we plan to monitor the molecular response of the PP arm as PCP1 engages with partner domains when relaxation will be a challenge regardless of these equilibria. We have clarified that minimizing relaxation is a desired feature in general, rather than an immediate need for isolated PCP1. PCP1 loaded with its PP arm and substrate is 10 kDa. We have expanded on these points in the introduction, sample preparation, and future directions sections (Sects. 1, 3.2, and 5) by adding the following text:

In Sect. 1: "We and others have found that some CPs interact transiently with their tethered substrates […] such that the phosphopantetheine group and its attached substrate sample both an undocked state and a docked state."

And later:

"The NMR linewidths of the tethered moiety indicate that the arm does not tumble independently from the protein core but is also not rigidly docked onto the protein, in line with a transient interaction."

And:

"Our immediate objective is to attenuate these residual signals and mitigate sensitivity losses for the targeted signals of unlabeled moieties, which will be particularly important for future studies of PCP1 engaging with its larger partner domains."

In Sect. 5: "Further experiments using these improved filters will enable studies of interactions between the prosthetic arm and PCP1, in isolation and in presence of its catalytic partner domains."

In Sect. 3.2: "Briefly, PCP1 (9.6 kDa) is expressed as a $His_6$-GB1 fusion protein containing a Tobacco Etch Virus (TEV) cleavage site."

"Upon confirmation of loading, purified Cys-loaded PCP1 (10 kDa with attached prosthetic group) was concentrated and buffer exchanged into NMR buffer containing 20 mM sodium phosphate pH 6.59 at 22 °C, 150 mM NaCl, 1 mM EDTA and 2 mM TCEP."

**2. Although the authors point out the value of being able to combine the WATERGATE suppression block with the third (delayed-decoupling) tuned filter element, they are perhaps missing a trick in that earlier schemes as reviewed and described in (Breeze, 2000) already featured incorporation of WATERGATE into the second filter element. This approach already shortens the scheme relative to one in which the WATERGATE block is sequentially positioned after the double-tuned filter, so in that sense the comparison they present with their 'reference experiment' should reflect this fact (granted, their new scheme still has an advantage in filtering efficiency by introducing the third element with complementary J tuning – but it will suffer a slight sensitivity loss compared with the experiment with doubly-tuned filter incorporating WATERGATE into the second element).**

Response: We are extremely grateful for this comment as it revealed wording that led to confusion, in particular as we did not mention that WATERGATE elements could be included in filters. Indeed, we

must modify our text to clarify what is compared with what and when, in particular when describing the advantages provided by our method. To assess the efficiency of our filter we needed a reference with the same pulse sequence length, as the filter would otherwise benefit from relaxation losses making the signals smaller not only due to the filter itself but also because of the duration of the filter. Thus, we started from a standard pulse sequence with a WATERGATE after the filters as a reference, and we incorporated our filter into that WATERGATE element. Here, when comparing our method with the reference, the attenuation of the residual protein core signals directly reports on the filter efficiency, without contamination by relaxation. However, when describing how our method improves on existing strategies, the "reference" (or point of comparison) does not need to be subject to this constraint, and indeed WATERGATE elements may already be combined with the last filters. To prevent confusion, we now avoid stating that we provide a filter without relaxation losses as this is only true if a stand-alone WATERGATE is available. Instead, we refer to a filter that mitigates relaxation losses through a shared evolution, or similar wording. While doing so, we also seize the opportunity to highlight that WATERGATE elements have already been incorporated within filters in previously published work. We apologize for this omission as we certainly knew we were not the first to do that.

We have added text in the manuscript to clarify the above points and credit the first uses of a repurposed WATERGATE – X half-filter as in (Breeze, 2000; Sattler, 1999):

*In Sect. 4*: "In reference experiments, the $X_d$ block is replaced by a 3-9-19 water suppression scheme, thus keeping all pulse sequences the same length for comparison. This consideration ensures that attenuations in signal intensities report exclusively on the efficiency of the filter and not on relaxation. The 3-9-19 scheme simply omits the inversion pulses on $^{13}C$ and $^{15}N$ shown in the $X_{d,J3}$ block, as well as the delayed composite pulse decoupling sequences."

"Incorporation of water suppression schemes in X-half filters has already been described (Breeze, 2000; Sattler et al., 1999). Briefly, inversion pulses are applied on $^{13}C$ and $^{15}N$ concomitantly with the existing proton inversion, here in the form of a 3-9-19 sequence, to enable evolution under scalar couplings. In our strategy, composite pulse decoupling is then delayed until coherences have become antiphase during detection." Including the new reference to:

Sattler, M., Schleucher, J. and Griesinger, C.: Heteronuclear multidimensional NMR experiments for the structure determination of proteins in solution employing pulsed field gradients, Prog. Nucl. Magn. Reson. Spectrosc., 34(2), 93–158, https://doi.org/10.1016/S0079-6565(98)00025-9, 1999.

*In Fig. 4 Caption*: "The pulse sequences used to obtain all spectra that are compared have the same lengths, and the comparisons report exclusively on the efficiency of the filters."

In all places where the wording was referring to general advantages of our method, we made sure to use terms such as "mitigate" or "minimize" when describing relaxation losses.

e.g:

Abstract: "[…] can be attenuated with mitigated sensitivity losses […]"

Sect. 1: "Our immediate objective is to attenuate these residual signals and mitigate sensitivity losses […]"

"[…] a method to attenuate undesired signals that escaped traditional filters with minimal increase in

the length of the pulse sequence."

"[…] to attenuate residual signals from coupled spins that have escaped filters with minimal or no increase in the lengths of pulse sequences"

Sect. 2:

"[…] at reduced costs in sensitivity for the signals of unlabeled moieties"

Sect. 4:

"[…], thus mitigating relaxation losses"

**3. 2 is not as clear as it might be. It's unclear (needs to be stated) (i) that these are simulations (ii) what the solid grey and dotted lines are, and what the value of tau is in every case (i.e. ratio to J)**

Response: We have updated the caption to Figure 2 to clarify the meaning of solid, grey, and dashed/dotted lines in the simulations. The new caption to Figure 2 is now:

"**Figure 2. Principles of editing through delayed decoupling.** (a) Applying decoupling once coherences are antiphase truncates their FID and attenuates their signals (dashed line), as shown here for the isolated component of a doublet. (b) The two components combine into a broadened and attenuated shape (dashed line). The analytical expressions of Eqs. (2) (solid grey line) and (4) (dashed black line) were used in (a) and (b). (c) Further attenuation is obtained when evolution into antiphase coherences is shared between a preparation period and detection as shown through simulations. The total evolution, $\Delta$, was set to $1/2J$, with evolutions during detection $\tau = 1/2J$ (dashed line), $1/4J$ (dotted line), and $1/8J$ (solid line). In (a)-(c), spectra without delayed decoupling are shown in grey for reference. (d) Simulation where the duration $\Delta$ is arrayed for a fixed preparation period $\Delta_{prep} = 1/4J$, and $\tau$ ranges from zero to $3/4J$ leading to $\Delta = 1/J$ in ten increments $\Delta\tau$ of $3/40$ J. This simulation predicts the results seen in Fig. 4(b). In (a)-(d), J is set to 120 Hz. (e) A delayed decoupling targeting 150 Hz leads to residual positive in-phase signals for spins with couplings at 120 Hz. (f) A delayed decoupling targeting 120 Hz leads to negative residual in-phase signals for couplings at 150 Hz. In (e) and (f), $\Delta_{prep} = 1/4J$ and $\tau$ is set to $1/4J$ for the targeted J, i.e. half of the total duration $\Delta$."

**4. Near the end of p5, it would be helpful to use consistent nomenclature to describe sinc function convolution (they use Sa function).**

Response: We have replaced the reference to a Sa function on page 5 with a sinc function for consistency:

"This description is reminiscent of discussions of truncation artefacts, which, in the frequency domain, lead to the convolution of Lorentzian signals with a sinc function. "

**5. Minor points to do with sample preparation: (i) why so much (presumably unlabelled) EDTA in a filtered experiment? (ii) Use of TCEP not advisable in phosphate buffer.**

Response: We thank the reviewer for pointing out that the concentration of EDTA was unnecessarily high. We will lower this concentration in future studies. Regarding TCEP, unfortunately, we cannot use DTT or similar thiols as reducing agents as we have observed disulfide-thiol exchange leading to DTT adducts on the PP arm and/or dimerization of cysteine-loaded PCP1 through disulfide bond formation. Addition of TCEP successfully restored the sample to the original form. Indeed, TCEP is much more rapidly oxidized in phosphate buffer than other buffers, and we buffer exchange our NMR samples with fresh buffer and degas the NMR tube before adding argon at the final stage of sample preparation. We use phosphate buffer because our first application will be to compare data of loaded PCP1 with that of apo-PCP1, which had been collected in phosphate buffer (Harden and Frueh, 2017).

---

## Author Response (AR2)

We thank the editor for the insightful clarifications and comments to our submission. Please see responses to each point below. Editor comments are in bold.

**1. Open data**

**Magnetic Resonance has quite a forward policy with respect to open data (https://www.magnetic-resonance-ampere.net/policies/data_policy.html). This aspect was key when MR was created and should be followed as much as possible. As a consequence, "available upon request" is not acceptable for MR.**

**The ideal case would be for you to post you matlab code, pulse sequence and parameters as well as raw NMR data in a well-managed open repository where these elements would be given DOIs. These DOIs would then be mentioned in the final version of your manuscript.**

**If this is not possible, the Matlab code and pulse sequence with parameters should be published alongside the article as supporting information.**

We have deposited all of the NMR data (including acquisition parameters, processing scripts, and pulse sequences), Matlab codes used for simulations, and MALDI-TOF data into the open repository Zenodo where it was assigned a DOI (http://doi.org/10.5281/zenodo.4730836). We have updated the Code and Data Availability section of the paper:

"**Code and Data Availability.** All NMR data including acquisition parameters, pulse sequences, and NMRPipe processing scripts, Matlab codes, and MALDI-TOF data are available for download from Zenodo at http://doi.org/10.5281/zenodo.4730836 (Marincin et al., 2021)."

A citation to the open-repository data has been included in the references:

"Marincin, K., Pal, I., and Frueh, D.: Using delayed decoupling to attenuate residual signals in editing filters [Data set], Zenodo, available at: http://doi.org/10.5281/zenodo.4730836, last access: 30 April 2021."

We have removed both instances where we have mentioned that simulation codes or pulse sequences are available upon request.

**2. Formatting**

**Equations are sometimes written in a surprising way: why is 2piJ/2 not written piJ? Also, 1/2J is understandable but ambiguous, the ideal would be 1/|2J| but 1/(2J) would already be much better. This should be done in the text and in Figure 2c.**

**In Figure panels 2e and 2f, there should be units for Delta, for instance as 1 s /(2·150).**

We thank the editor for clarifying this formatting issue. We have simplified the $2\pi J/2$ on page 5 to:

"i.e. for a two-spin system on resonance $\omega_0 = +/- \pi J$, and R is a transverse relaxation…"

and on page 8 to:

"…the simulation is only performed on resonance such that $\omega_{0\alpha} = \pi J$ and $\omega_{0\beta} = -\pi J$."

To answer your question: the notation $2\pi J/2$ simply reflects that the P.I. did too much teaching, when he delineates the conversion to rad/s ($2\pi$) from the frequency offset for a 2-spin system (+/- J/2 when on resonance).

We have also implemented the suggested notation for every instance of fractions that involve scalar-coupling:

"…in which decoupling is applied after a time $\tau = 1/|2J|$ and assume…" (p. 5)

"…in the example we discussed, $\tau$ is set to $1/|2J|$, our objective is to…" (p. 7)

"…applying decoupling is kept at $\Delta = \Delta_{prep} + \tau = 1/|2J|$, where $\Delta_{prep}$ is…" (p. 7)

"…$\tau$ takes the values $1/|2J|$ (when $\Delta_{prep}$ is zero), $1/|4J|$, and $1/|8J|$." (p. 7)

"…when decoupling is applied before reaching $1/|2J|$, a residual positive in-phase signal is detected, whereas a negative in-phase signal emerges passed $1/|2J|$." (p. 7)

"…when $\Delta$ exceeds the optimal value of $1/|2J|$." (p. 7)

"…on resonance for delays $\Delta$ slightly exceeding $1/|2J|$, …" (p. 7)

"…the value of $\tau$ selected through visual inspection typically exceeds $1/|2J|$ as signal suppression appears more efficient at those values than at $1/|2J|$." (p. 7)

"…that the length of each block in our pulse sequence is maintained to $1/|2J(NH)|$ so as to permit…" (p. 12)

The figure has been updated to include units in panels 2(e) and 2(f), as well as correcting the fractional J-coupling notation used on panel 2(c) (see updated figure in latest paper version). The caption to Figure 2 has been updated to:

"**Figure 2. Principles of editing through delayed decoupling.** (a) Applying decoupling once coherences are antiphase truncates their FID and attenuates their signals (dashed line), as shown here for the isolated component of a doublet. (b) The two components

combine into a broadened and attenuated shape (dashed line). The analytical expressions of Eqs. (2) (solid grey line) and (4) (dashed black line) were used in (a) and (b). (c) Further attenuation is obtained when evolution into antiphase coherences is shared between a preparation period and detection as shown through simulations. The total evolution, $\Delta$, was set to $1/|2J|$, with evolutions during detection $\tau = 1/|2J|$ (dashed line), $1/|4J|$ (dotted line), and $1/|8J|$ (solid line). In (a)-(c), spectra without delayed decoupling are shown in grey for reference. (d) Simulation where the duration $\Delta$ is arrayed for a fixed preparation period $\Delta_{prep} = 1/|4J|$, and $\tau$ ranges from zero to $3/|4J|$ leading to $\Delta = 1/|J|$ in ten increments $\Delta\tau$ of $3/|40J|$. This simulation predicts the results seen in Fig. 4(b). In (a)-(d), J is set to 120 Hz. (e) A delayed decoupling targeting 150 Hz leads to residual positive in-phase signals for spins with couplings at 120 Hz. (f) A delayed decoupling targeting 120 Hz leads to negative residual in-phase signals for couplings at 150 Hz. In (e) and (f), $\Delta_{prep} = 1/|4J|$ and $\tau$ is set to $1/|4J|$ for the targeted J, i.e. half of the total duration $\Delta$."

The caption to Figure 3 has been updated to include:

"The delays in the $X_{J1}$ and $X_{J2}$ filter blocks are: $\delta_3 = 1/|4\,J(NH)| \approx 2.78$ ms, $\delta_1 = 1/|4\,J1(CH)|$, $\delta_4 = 1/|4\,J2(CH)|$, $\delta_2 = 1/|4\,J(NH)| - 1/|4\,J1(CH)|$, and $\delta_5 = 1/|4\,J(NH)| - 1/|4\,J2(CH)|$."

**On line 233, I believe 10.008 Hz should be 10.008 kHz.**

We thank the editor for catching this mistake. We have corrected this sentence to:

"The field strength of the DIPSI-2 TOCSY mixing sequence was 10.008 kHz and…"

**3. Two points worthy of a discussion?**

**On page 9, you mention that 7 Hz line broadening was used. Given the expected line shape for IS pairs with scalar couplings above the target scalar coupling (i.e. Figure 2f), I wonder how much of an effect this can have by smoothing out these residual peaks.**

You are right to highlight that 4(b) differs from 2(d) (and 2(e) or 2(f)) in part due to apodization, the other effect being overlap. Note that all signals report on CH systems with about the same scalar coupling, and hence optimal attenuation is obtained for the same value of $\tau$, which leads to the spectrum in black in figure 4(b). At this value all residual signals appear as the orange curve in 2(f) (or the blue curve in 2(e)) and there are no residual sharp peaks. However, as you rightly pointed out, the appearance of the array is indeed different from that in 2(d) due to smoothing. We added a sentence to attract the attention of the reader to those aspects.

"The differences between the line shapes of Figs. 2(d) and 4(b) reflect signal overlap and apodization."

We have also updated the text in Section 3.3 to expand on the use of line broadening:

"All 1D spectra were zero-filled to 4096 points before Fourier transform and subsequently apodized using exponential multiplication with 7 Hz broadening to reduce truncation artefacts from buffer signals."

**You present here the use of delayed decoupling in a fully filtered experiment, which is a perfectly legitimate and useful experiment. However, isotope filters are often run to detect intermolecular transient NOEs in experiments that are part isotope filtered and part isotope edited. In the common case where the isotope filter is performed before the NOE mixing time, I wonder if and how delayed decoupling could be used for an evolution in an indirect time dimension instead of detection.**

We thank the editor for this insightful comment. Indeed, our immediate objective is to improve filtering methods in the detected dimension, notably to get 1D filtered spectra, and we focus exclusively on this objective in the current manuscript. We do have a solution to incorporate delayed decoupling in indirect dimensions. However, we have not yet tested this solution. Further, some methods already exploit evolutions during indirect dimensions to improve filtering, and we have not tested whether delayed decoupling would even be needed for these experiments.

We have included language to emphasize our focus on filtering in the detected dimension. For example:

"Many methodologies have been implemented to filter signals from labeled molecules in direct or indirect dimensions, reviewed in (Breeze, 2000; Robertson et al., 2009). As our immediate objective is to obtain 1D proton spectra of unlabeled moieties, we do not consider methods that exploit evolutions in indirect dimensions and here, we focus solely on filters for the detected dimension." (p. 4)

"Here, we present a method to attenuate undesired signals in the detected dimension that escaped traditional filters with minimal increase in the length of the pulse sequence."
(p. 4)

"Inclusion of delayed decoupling in the detected dimension of a sequential tuned filter removes spurious signals with minimal costs in sensitivity…" (p. 15)

"As an immediate application, we show enhanced suppression of undesired PCP1 signals when $X_d$ is incorporated into the direct dimension of a sequential X-half-filtered 2D TOCSY experiment (Fig. 5)." (p. 15)